# Voluntary Surveillance Program for Equine Influenza Virus in the United States during 2008–2021

**DOI:** 10.3390/pathogens12020192

**Published:** 2023-01-27

**Authors:** Duane E. Chappell, D. Craig Barnett, Kaitlyn James, Bryant Craig, Fairfield Bain, Earl Gaughan, Chrissie Schneider, Wendy Vaala, Samantha M. Barnum, Nicola Pusterla

**Affiliations:** 1Merck Animal Health, Madison, NJ 07940, USA; 2Department of Medicine and Epidemiology, School of Veterinary Medicine, University of California, Davis, CA 95616, USA

**Keywords:** equine influenza virus, upper respiratory tract infection, qPCR, equids, viruses and bacteria, prevalence factors, respiratory tract

## Abstract

A voluntary upper respiratory biosurveillance program in the USA received 9740 nasal swab submissions during the years 2008–2021 from 333 veterinarians and veterinary clinics. The nasal swabs were submitted for qPCR testing for six common upper respiratory pathogens:equine influenza virus (EIV), equine herpesvirus-1 (EHV-1), equine herpesvirus-4 (EHV-4), *Streptococcus equi* subspecies *equi* (*S. equi*), equine rhinitis A virus (ERAV), and equine rhinitis B virus (ERBV). Additional testing was performed for equine gamma herpesvirus-2 (EHV-2) and equine gamma herpesvirus-5 (EHV-5) and the results are reported. Basic frequency statistics and multivariate logistic regression models were utilized to determine the associations between risk factors and EIV positivity. The EIV qPCR-positivity rate was 9.9%. Equids less than 9 years of age with a recent history of travel and seasonal occurrence in winter and spring were the most common population that were qPCR positive for EIV. This ongoing biosurveillance program emphasizes the need for molecular testing for pathogen identification, which is critical for decisions associated with therapeutics and biosecurity intervention for health management and vaccine evaluations and development.

## 1. Introduction

Equine influenza virus (EIV) belongs to the Orthomyxoviridae family of negative-stranded RNA viruses [1]. Two distinct subtypes have been isolated since 1956, Influenza A/equine/Prague/56 (H7N7) and Influenza A/equine/Miami/63 (H3N8) [2]. Prague ’56 (H7N7) has not been isolated in horses since the 1970s [2,3]. The Miami ’63 strain diverged into Eurasian and American lineages, named according to their geographic regions [2,3]. The Eurasian lineage has not been detected since 2005 and the American lineage evolved into Kentucky, South American, and Florida sublineages [2,3]. Florida sublineages have become the dominant strain and further drifted into clade 1 and 2 subgroups, with clade 1 reported in North America and clades 1 and 2 in Europe and other parts of the world [2,3,4,5].

Equine influenza (EI) continues to be an international enzootic cause of infectious respiratory disease in equids, except for Australia, New Zealand, and Iceland [2,3,4,6]. With near worldwide distribution, this highly contagious and rapidly spreading infection by direct and indirect contact has led to multiple outbreaks in non-vaccinated and vaccinated individuals [3]. Common clinical signs include fever, lethargy, nasal discharge, and a non-productive cough [2,3]. Horses will usually recover in 2 weeks, although a cough may persist for longer periods of time [2,3]. Seasonal occurrence is most often observed in winter and spring, although cases have been documented throughout the year [7]. Morbidity rates can be quite high, coupled with low mortality; a noted exception is the donkey population who experience both high morbidity and mortality rates [2,3]. Although loss of life can occur, economic loss is the most impactful, with extended periods of rest required for complete recovery in both performance and working-class equids around the world [3].

Ongoing management efforts through vaccination, identification, and appropriate quarantine are all necessary to maintain herd health against this challenging virus. This manuscript investigates the prevalence and epidemiology of EI in equids from March 2008 to June 2021 obtained from voluntary nasal swab and whole blood sample submissions by veterinarians across the United States of America. This ongoing voluntary surveillance program confirms that EIV continues to be one of the most common respiratory pathogens of the horse, equal to or a close second to equine herpesvirus type-4 (EHV-4).

## 2. Materials and Methods

### 2.1. Sample Collection, Handling and Processing

Veterinarians from 333 clinics representing 42 states provided nasal swabs (Puritan Products Company LLC, Guilford, ME, USA) and whole blood samples (Becton Dickinson, Franklin Lakes, NJ, USA) [6,8,9] to an ongoing equine upper respiratory pathogen surveillance program. The minimum criteria for sample submission included fever (defined as a rectal temperature > 101.5 °F) (38.6 °C) and one or more of the following clinical signs: nasal discharge, cough, lethargy, and/or central nervous system signs. Case submission occurred from March 2008 to June 2021. Portions of this dataset have been reviewed in previous publications [6,7,9].

### 2.2. Insert Questionnaire

Each submission was accompanied by a questionnaire (see Appendix A) capturing signalment, vaccination and travel/exposure history, and clinical signs. The signalment included age, sex, breed, occupation/use (racing, show, pleasure, breeding, other), vaccination history (product last used, date last vaccinated, and number of doses per year) and travel/exposure history (number of days showing clinical signs, transport in last 14 days, and number of horses on the premises). Clinical signs that were reported included temperature at the time of sample collection and the presence of a cough, nasal discharge, lethargy, central nervous system signs, limb swelling, loss of appetite, and ocular discharge, with graded responses of none observed, mild, moderate, or severe for each clinical sign.

Two nasal swab samples were collected from the nostril with the most obvious nasal discharge with 6” plastic handle sterile rayon-tipped swabs (Puritan Medical Products LLC, Guiliford, Maine). The swabs were placed in viral transport media (2 mL containing 0.125% gentamicin and 0.1% amphotericin B) or in a sterile red top tube (Becton Dickinson, Franklin Lakes, NJ, USA). A 3–5 mL blood sample was requested in an EDTA purple top blood tube from each horse. The samples were refrigerated until shipping and veterinarians were encouraged to ship the samples with an ice pack overnight to the diagnostic laboratory for qPCR testing. The samples were tested within 7 days of collection.

Nucleic acid extraction from nasal secretions and EDTA blood was performed the day of sample arrival to the laboratory using an automated nucleic acid extraction system according to the manufacturer’s recommendations (QIAcubeHT, Germantown, MD, USA). The Quantitect Reverse transcription kit (Qiagen) was used for cDNA synthesis following the manufacturer’s directions with the following modifications. A volume of 10 μL of RNA was digested with 2 μL of gDNA WipeOut Buffer by incubation at 42 °C for three minutes and was then briefly centrifuged. Then, 8 μL Quantitect Reverse Transcriptase were added and brought up to a final volume of 20 μL and incubated at 42 °C for 40 min. The samples were inactivated at 95 °C for 3 min, chilled, and 60 μL of nuclease-free water was added to dilute the cDNA to an optimal concentration.

The total RNA was purified from nasal secretions and transcribed to complementary DNA, as previously described [9]. To determine the sample quality and efficiency of nucleic acid extraction, all of the samples were assessed for the presence of the housekeeping gene eGAPDH, as previously described [10]. Nasal secretions were assayed for the presence of the HA1 gene of EIV using a previously reported qPCR assay [6,9]. All of the nasal samples were also screened for equine herpesvirus-1 (EHV-1), equine herpesvirus-4 (EHV-4), and *Streptococcus equi* subsp. *equi* (*S. equi*). From September 2012 onwards, equine rhinitis A & B viruses (ERAV and ERBV) and equine herpesvirus-2 and equine herpesvirus-5 (EHV-2 and EHV-5) were also included in the nasal swab testing. DNA purified from the whole blood was screened for EHV-1, as previously reported [10].

### 2.3. Statistical Analysis

Demographic and clinical factors were compared by qPCR EIV-negative or -positive status. Demographic factors included breed, use, sex, age (analyzed continuously and categorized into five-year increments), history of transport, number of affected horses on the property, and days clinical signs present (categorized as 3 days or less, 4–7 days, >7 days). Vaccination history (EHV-1/-4, EIV, and *S. equi*) and the presence (of any severity) of nasal discharge, ocular signs, cough, limb edema, anorexia, lethargy, and CNS signs were also compared. Parametric (Chi-square and Student’s *t*-test) and non-parametric tests (Fisher’s exact and Mann–Whitney U test) were used, as appropriate, to compare categorical and continuous factors.

Unadjusted and adjusted logistic regression was used to model the association between qPCR EIV-negative vs.-positive status and the prevalence factors. Factors included in the final adjusted model were chosen a priori, and included year, season, age, breed, history of transport, number of horses affected on the property, and a composite of any respiratory sign (including ocular signs, nasal discharge, and/or cough). To further explore the relationships between prevalence factors and qPCR EIV-negative or -positive status, secondary models including sex and use were also generated. The results are reported as prevalence odds ratios and 95% confidence intervals. For all statistical analyses, values of *p* ≤ 0.05 were considered significant. All of the analyses were conducted in StataIC, version 16.0.3 (Stata Statistical Software, College Station, TX, USA).

## 3. Results

Here, 9740 horses were sampled with nasal swabs and EDTA blood submissions from March 2008 to June 2021. There were 966 samples that tested positive for EIV through qPCR with a positivity rate of 9.9%. (Table 1). There was an increasing trend when evaluating time as a continuous factor from 2008 to 2021 (Figure 1). The years of highest prevalence of EIV-positive sampling were 2019 (15.3% incidence), 2013 (12.5%), and 2020 (12.4%) (Figure 1). 

Samples were submitted from 42 states across the USA, with EIV qPCR-positive samples received from 35 states (Alabama, Arizona, California, Colorado, Florida, Georgia, Iowa, Idaho, Indiana, Kentucky, Louisiana, Maryland, Michigan, Minnesota, Missouri, Montana, North Carolina, Nebraska, New Jersey, Nevada, New York, Ohio, Oklahoma, Oregon, Pennsylvania, Rhode Island, South Carolina, South Dakota, Tennessee, Texas, Utah, Virginia, Washington, Wisconsin, and West Virginia). EIV-negative swabs originated from Connecticut, Delaware, Illinois, Kansas, Mississippi, Vermont, and Wyoming.

Quarter Horses (QH) more frequently tested EIV qPCR positive than other breeds (Table 1). Based on a univariate logistic regression (Table 2), Thoroughbreds (TB), Warmbloods (WB), and Arabians were less likely to test EIV qPCR positive compared with QH (*p* < 0.001). It was also noted that submissions from draft horse breeds were more likely to be EIV qPCR positive compared with QH submissions in a multivariate model with OR 1.62 (95% CI 1.6, 2.46, *p* < 0.01).

No significant difference was observed between EIV qPCR-positive and EIV qPCR-negative outcomes with respect to the horse’s use or occupation at the time of sample submission. The samples were received from a wide range of occupations identified as either competition, pleasure, breeding, other, or unknown (meaning not indicated on the submission form).

Mares were more likely to be EIV qPCR positive than geldings/stallions (*p* < 0.009) (Table 1). When sex and use were included in a logistic regression model to evaluate if sex selection was influenced by occupation or use, the outcome still showed gelding/stallion submissions were less likely to be EIV qPCR positive compared with mares (*p* = 0.007, OR 0.81 (95% CI 0.70, 0.94).

Regarding the variable of age, using less than one year as a point of reference (Table 2), both age groups 1 to 4 years and 5 to 9 years had double the odds of a qPCR-positive EIV result (aOR 2.04 (95% CI 1.55, 2.67) *p* < 0.001 and aOR 2.28 (95% CI 1.72, 3.00) *p* < 0.001, respectively). The age groups (Figure 2) of 10–14 years and 15–19 years (OR of 1.37 and 0.96, respectively) were less significant and displayed a similar likelihood of EIV qPCR-positive vs. EIV qPCR-negative outcome. Horses older than 20 years of age had an OR of 0.42, displaying only half the likelihood of a qPCR-positive EIV sample.

The impact of recent transportation within the past 14 days prior to sampling showed a significant impact on EIV qPCR positivity in the final adjusted model (aOR 1.76 (95% CI 1.48, 2.09); *p* < 0.001) (Table 2).

As the study primarily enrolled sentinel cases, the impact of herd size displayed a profound effect on EIV qPCR-positivity rate in the final adjusted model, with an aOR of 3.08 (95% CI 2.59, 3.64) *p* < 0.001 (Table 2) for samples coming from groups of more than one horse affected.

EIV qPCR-positive horses were more likely to have clinical signs lasting seven days or less from the time of sample submission. The overall global *p*-value is <0.001; in further post hoc pairwise comparison testing, EIV qPCR-negative horses were more likely to be diagnosed with more than 7 days of symptoms (*p* < 0.05), EIV qPCR-positive horses were more likely diagnosed while displaying clinical signs for 4–7 days (*p* < 0.05); no difference in EIV qPCR-negative or -positive status when looking at horses with 3 or less days of clinical signs being present (Table 1).

Evaluation of the effect of season of the year using winter as the point of reference (Table 2/Figure 3) determined that the highest prevalence of EIV qPCR-positive samples occurred during the winter/spring months (Dec–May). The adjusted model displayed a more likely EIV qPCR-negative outcome in summer/fall seasons with aOR 0.47 (95% CI 0.37, 0.59) and 0.49 (95% CI 0.40, 0.63), respectively.

Evaluation of vaccination history (Table 3), in cases where EIV vaccination status was known, revealed a significant difference in EIV qPCR-positive cases (23.3%) compared with EIV qPCR-negative cases (34.1%) with *p* < 0.001. When looking at EHV-1/4 vaccination, which is often coupled with EIV in a multi-antigen vaccine, similar impact was seen with a greater incidence of EIV qPCR-negative outcomes (34.5%) vs. EIV qPCR-positive outcomes (23.3%) with *p* <0.001. The most reported vaccine information was “unknown” in over 50% of cases and yielded a higher rate of EIV qPCR-positive samples vs. EIV qPCR-negative samples.

Clinical signs of EI (Table 3) are hallmarked by acute fever, nasal discharge, and cough. All were displayed in this surveillance program with *p*-values <0.001. In addition, significant *p*-values (<0.001) were also noted with ocular signs, limb edema (although less often reported), and lethargy. It should be noted that the submission criteria to qualify for this surveillance program were temperature > 101.5 °F (38.6 °C), and one of the following clinical signs: nasal discharge, cough, lethargy, and/or central nervous system deficits. In the multivariate model (Table 2), a composite of the presence of any respiratory sign (nasal discharge, ocular discharge, and/or cough) showed a high occurrence in EIV qPCR-positive cases (aOR 10.46 (95% CI 6.5, 16.8) *p* < 0.001).

EIV qPCR-positive cases were less likely to be co-infected with EHV-1 (1.9% EIV qPCR negative vs. 0.3% EIV qPCR positive, *p* < 0.001) (Table 3). EIV qPCR-positive cases were less likely to be co-infected with EHV-4 (10.9% EIV qPCR-negative vs. 6.2% EIV qPCR positive, *p* < 0.001). EIV qPCR-positive cases were less likely to be co-infected with *S. equi* (8.1% EIV qPCR negative vs. 3.1% EIV qPCR positive, *p* < 0.001). ERAV qPCR-positive cases did not show significant association with EIV qPCR-negative or -positive cases (0.1% EIV qPCR negative vs. 0.1% EIV qPCR positive, *p* = 0.73). ERBV qPCR-positive cases did show a significant association with EIV qPCR-negative or -positive cases (3.7% EIV qPCR negative vs. 2.1% qPCR positive, *p* = 0.003). A comparison of the prevalence of EHV-2 and EHV-5 co-infection with EIV qPCR-positive vs. EIV qPCR-negative cases did not show a significant difference. The prevalence of EHV-2 and EHV-5-positive samples was approximately 40% with or without co-infection with EIV qPCR-positive cases. A significant difference in co-infection rate was noted in the < 1-year age group, where 16.6% of EIV qPCR negative vs. 7.4% of EIV qPCR positive vs. 13.0% EIV qPCR positive and co-infection with at least one additional pathogen were noted (*p* < 0.001) (Table 4). In addition, a significantly noted difference in co-infection rate was observed in the < l year age group comparing EIV qPCR negative (16.6%) to EIV qPCR positive co-infected with EHV-2 qPCR positive (14.7%) cases (*p* < 0.001) (Table 5). No other age groups displayed significant differences in co-infection rates.

## 4. Discussion

This ongoing voluntary biosurveillance study identifying the most common upper respiratory tract pathogens in the equid population continues to serve the equine industry with four primary goals: assist with accurate and timely diagnostic answers to acute respiratory challenges, provide a better understanding of the prevalence and epidemiology of equine infectious respiratory pathogens, identify and monitor current circulating strains of major respiratory pathogens, and evaluate the efficacy of current vaccination protocols. EIV continues to be identified as one of the most common upper respiratory pathogens often running neck and neck with EHV-4.

This study population represents a sampling of a sentinel population with acute upper respiratory illness, which affords a greater opportunity for pathogen identification, one of the four goals of this program. Qualifying criteria for enrollment included fever (temperature greater than 101.5 °F (38.6 °C)) and the presence of one of the following clinical signs: nasal discharge, cough, lethargy, and/or central nervous system signs. The diagnostic information generated allows for both improved management of a respiratory disease outbreak and retrospective review for future decision-making and learning.

Sample submission from a vast majority of the states within the continental United States have identified similar trends such as seasonal occurrences of equine influenza most often in winter and spring [3]. However, it should not be overlooked that EIV infection can occur and be identified in any month of the year. The seasonality of EIV infection should stimulate re-evaluation of the timing of routine EI vaccination. As an alternative to biannual EIV vaccination schedules, one could shift August/September booster vaccinations to November/December [7] or continue with bi-annual vaccinations in spring and early fall with the addition of a third EIV booster vaccination in November/December. The suggested addition of a third EIV booster would be for unique situations for horses experiencing a high risk of exposure and great perceived loss in training and preparation for winter events.

In this biosurveillance program, the Quarter Horse breed, representing the largest breed registry in the United States [11], contributed a large portion of the samples in the submission pool. This fact is not unexpected, but the adjusted regression model reinforces the finding that the occurrence of EIV qPCR-positive sampling was more frequent in this breed compared with other breeds sampled. This observation may be impacted by study participants’ veterinary clinical practice demographics and the multifaceted use of the Quarter Horse breed in many different disciplines.

Occupation or use of the horse, as described on the submission form, did not impact the EIV qPCR-positivity rate. The sex of the sampled horse was significant, with mares more commonly being EIV qPCR positive compared with geldings or stallions. There may be no correlation, but a similar observation noted that mares were found to be at higher risk of EHV-1 infection in the retrospective study of the 2011 EHV-1 outbreak in Ogden, Utah, USA [12]. 

EIV is commonly referenced as a young horse disease [1]. This study identifies a broader age range than previously expected, with the highest rate of EIV qPCR-positivity in age groups of 1–4 years of age and 5–9 years of age when compared with <1 year of age [13]. Equids greater than 9 years old show a declining pattern of EIV incidence with increasing age. Equids older than 20 years are much less likely to have an EIV qPCR-positive sample. This age comparison is similarly identified in The World Organization for Animal Health (Office International des Epizooties, or OIE) findings [13]. The identification of a decreasing incidence of EI in aging horses in this study is contrary to information previously published identifying a reduction in anamnestic response to EI vaccination with aging [14].

Travel has been shown to impact disease prevalence due to stress factors and the influence on immune responses, co-mingling of horses at show/exhibit/breeding/sale locations, and varying levels of biosecurity between one event to another [7]. 

Seasonality of EIV infection is clearly defined in winter and spring months, as seen throughout this dataset review, and supports prior observations in smaller subsets of this data [6,8,9]. It is important to note that EIV qPCR-positive samples continue to be identified in the summer and fall. This fact, combined with the 6-month duration of immunity of many EIV vaccines in the United States, supports the recommendation for biannual vaccination, as suggested by the American Association of Equine Practitioners (AAEP) vaccination guidelines for risk-based diseases [15]. Vaccination history was requested on the submission form for this study and would have been helpful for retrospective analysis. Unfortunately, in over 50% of the sample submissions, this data was not available. This lack of information leaves many unanswered questions relating to the impact vaccination has on not only prevention, but on the duration of immunity with EIV vaccines. The vaccination information that was provided in this study continues to support the benefit of a bi-annual EIV vaccination program [1,16]. The vaccine should be relevant to circulating influenza strains and at minimum meet current OIE recommendations to contain both clade 1 (A/eq/Ohio/2003-like) and clade 2 (A/eq/Richmond/1/2007-like) viruses of the Florida sublineage [4,6,13].

Clinical signs of respiratory disease (fever, nasal and ocular discharge, cough, and lethargy) often look similar across cases caused by various respiratory pathogens [17]. Identification of the causative pathogen is not possible based solely on clinical signs. Timely determination of the causative pathogen via molecular diagnostic testing can have a substantial impact on treatment recommendations and biosecurity measures [3].

The collection of nasal swab samples early in the course of disease (within 3 days of appearance of clinical signs) can lead to a greater opportunity to identify EIV versus waiting until clinical signs have been present for more than seven days when pathogen identification will be less likely. To note, the EIV incubation period is 1–3 days and the shedding period can be up to 10 days [1,5].

Monitoring for co-infection with other diseases is important as it has not been well-established if more than one pathogen is likely to be present along with EIV. Multi-pathogen respiratory disease occurs in other species, such as canine infectious tracheobronchitis (“kennel cough”) [18] and bovine respiratory disease complex in cattle [19], and there is a need to remain vigilant to determine if a similar phenomenon occurs in horses. Although the gamma herpesviruses are ubiquitous in the equine population and co-existence with a major pathogen such as EIV may be identified, the relationship between the pathogens needs further research. The observation in this study of co-infection with EHV-2 in horses less than 1 year of age may be explained by their developing immune system, decreasing maternal antibody protection, and varying vaccination protocols that may or may not be providing the desired immune responses.

This ongoing biosurveillance program will continue to open new doors of exploration, spur more questions into the causes of and risk factors for equine respiratory disease, and provide timely information to practitioners managing cases of acute respiratory disease in their equine patients. The program aims to guide the equine industry with accurate information regarding disease incidence, the impact of multiple demographic variables, and tools to assess vaccination and biosecurity protocols. Further study identifying the genetic differences of circulating EIV strains [1,20,21] will be crucial to evaluate differences in the currently available EIV vaccines versus field EIV strains. These supplemental genetic evaluations will provide organizations such as the OIE with the information needed for annual EIV vaccine review and recommendations [3].

## Figures and Tables

**Figure 1 pathogens-12-00192-f001:**
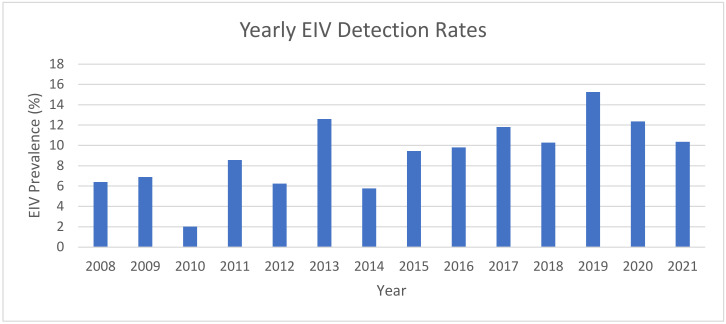
Percentage EIV prevalence compared with all of the samples submitted by year displaying yearly trend based on adjusted logistic regression (aOR 1.05 (95% CI 1.02, 1.08); *p* < 0.001).

**Figure 2 pathogens-12-00192-f002:**
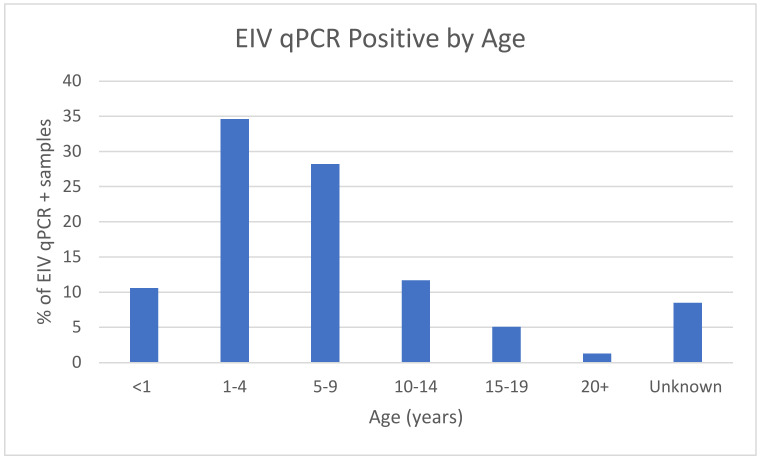
Percentage of EIV-positive samples by age.

**Figure 3 pathogens-12-00192-f003:**
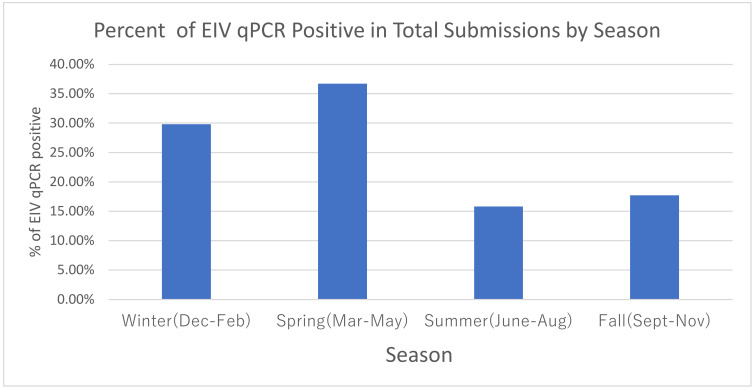
Seasonal impact on EIV qPCR positivity.

**Table 1 pathogens-12-00192-t001:** Demographic factors associated with EIV among 9740 horses in the USA from 2008–2021.

	EIV qPCR Negative (n = 8774)	EIV qPCR Positive (n = 966)	*p*-Value
**Breed**			
Quarter horse (QH)	3083 (35.1%)	404 (41.8%)	<0.001
Thoroughbred (TB)	1472 (16.8%)	84 (8.7%)	
Warmblood (WB)	1006 (11.5%)	64 (6.6%)	
Paint	394 (4.5%)	51 (5.3%)	
Arabian	584 (6.7%)	37 (3.8%)	
Draft	213 (2.4%)	38 (3.9%)	
Pony	330 (3.8%)	43 (4.5%)	
Other	1692 (19.3%)	245 (25.4%)	
**Use**			
Competition	3640 (41.5%)	379 (39.2%)	0.042
Pleasure	3235 (36.9%)	375 (38.8%)	
Breeding	376 (4.3%)	34 (3.5%)	
Other	743 (8.5%)	59 (6.1%)	
Unknown	780 (8.9%)	119 (12.3%)	
**Sex**			
Mare	3054 (34.8%)	381 (39.4%)	0.009
Gelding/Stallion	4415 (50.3%)	439 (45.4%)	
Unknown	1305 (14.9%)	146 (15.1%)	
**Age**			
<1	1454 (16.6%)	102 (10.6%)	<0.001
1–4	2111 (24.1%)	334 (34.6%)	
5–9	1803 (20.5%)	272 (28.2%)	
10–14	1355 (15.4%)	113 (11.7%)	
15–19	831 (9.5%)	49 (5.1%)	
20+	590 (6.7%)	13 (1.3%)	
Unknown	630 (7.2%)	83 (8.6%)	
**History of Transport**			
No	5751 (65.5%)	512 (53.0%)	<0.001
Yes	2261 (25.8%)	364 (37.7%)	
Unknown	762 (8.7%)	90 (9.3%)	
**Affected horses on Property**			
Single	6026 (68.7%)	425 (44.0%)	<0.001
Multiple	1960 (22.3%)	445 (46.1%)	
Unknown	788 (9.0%)	96 (9.9%)	
**Days Clinical Signs Present**			
3 days or less	5685 (64.8%)	620 (64.2%)	<0.001
4–7 days	2325 (26.5%)	317 (32.8%)	
>7 days	764 (8.7%)	29 (3.0%)	

**Table 2 pathogens-12-00192-t002:** Multivariate model associated with EIV among 9740 horses in the USA from 2008–2021.

	Adjusted OR (95% CI), *p*-Value
**Season**	
Winter	Ref
Spring	0.98 (0.81, 1.20); 0.88
Summer	0.47 (0.37, 0.59) <0.001
Fall	0.49 (0.40, 0.63) <0.001
**Age Category**	
<1 year	Ref
1–4 years	2.04 (1.55, 2.67) <0.001
5–9 years	2.28 (1.72, 3.00) <0.001
10–14 years	1.37 (0.99, 1.87) 0.055
15–19 years	0.96 (0.65, 1.42) 0.85
20+ years	0.42 (0.23, 0.78) 0.006
**Breed**	
QH	Ref
TB	0.45 (0.33, 0.61) <0.001
WB	0.73 (0.54, 0.99) 0.04
Paint	1.12 (0.77, 1.64) 0.56
Arabian	0.60 (0.41, 0.89) 0.01
Draft	1.62 (1.06, 2.46) 0.01
Pony	0.99 (0.68, 1.47) 0.99
Other	1.40 (1.13, 1.71) 0.002
**History of Transport**	1.76 (1.48, 2.09) <0.001
**Multiple Affected Horses on Property**	3.08 (2.59, 3.64) <0.001
**Respiratory Signs (Cough, Nasal, Ocular)**	10.46 (6.5, 16.8); <0.001
**Year**	1.05 (1.02, 1.08); <0.001

**Table 3 pathogens-12-00192-t003:** Vaccination, co-infection, and clinical signs associated with EIV detection from nasal swabs among 9740 United States horses, from 2008–2021.

	EIV qPCR Negative (n = 8774)	EIV qPCR Positive (n = 966)	*p*-Value
**Vaccine History**			
EHV-1/-4	3024 (34.5%)	225 (23.3%)	<0.001
Unknown EHV-1/4 history	4939 (56.3%)	600 (62.1%)	<0.001
EIV	2989 (34.1%)	225 (23.3%)	<0.001
Unknown EIV history	5006 (57.1%)	610 (63.1%)	<0.001
*S. equi* subspecies *equi*	854 (9.7%)	53 (5.5%)	<0.001
Unknown *S. equi* subspecies *equi* history	5094 (58.1%)	626 (64.8%)	<0.001
**qPCR positive Nasal Swab** **co-Infection**			
EHV-1 positive	168 (1.9%)	3 (0.3%)	<0.001
EHV-4 positive	954 (10.9%)	60 (6.2%)	<0.001
*S. equi* subspecies *equi* positive	708 (8.1%)	30 (3.1%)	<0.001
ERAV positive	12 (0.1%)	1(0.1%)	0.73
ERBV positive	322(3.7%)	20 (2.1%)	0.003
EHV-2 positive	3291 (37.5%)	381 (39.4%)	0.048
EHV-5 positive	3215 (36.6%)	385 (39.9%)	0.89
**Clinical Signs** (Presence, Any Severity)			
Nasal discharge	5985 (68.2%)	867 (89.8%)	<0.001
Ocular signs	1957 (22.3%)	257 (26.6%)	<0.001
Cough	3592 (40.9%)	811 (84.0%)	<0.001
Limb edema	887 (10.1%)	32 (3.3%)	<0.001
Anorexia	4932 (56.2%)	537 (55.6%)	0.056
Lethargy	5950 (67.8%)	678 (70.2%)	<0.001
CNS signs	593 (6.8%)	16 (1.7%)	<0.001

**Table 4 pathogens-12-00192-t004:** Demographic factors associated with EIV qPCR-defined nasal swab positivity with any coinfection.

Age	EIV qPCR Negative (n = 8774)	EIV qPCR Positive, No Coinfection (n = 418)	EIV qPCR Positive with any Coinfection (n = 548)	*p*-Value
<1 year of age	1454 (16.6%)	31 (7.4%)	71 (13.0%)	<0.001

**Table 5 pathogens-12-00192-t005:** Co-infection comparison of EIV qPCR negative to EIV qPCR positive with or without EHV-2 co-infection (2012–2021).

Age	EIV qPCR Negative (n = 6607)	EIV qPCR Positive, EHV-2 qPCR Negative (n = 405)	EIV qPCR Positive, EHV-2 qPCR Positive (n = 381)	*p*-Value
<1 year of age	1094 (16.6%)	22 (5.4%)	56 (14.7%)	<0.001

## Data Availability

Data available upon request due to privacy restrictions.

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
