# Peer review of "Voluntary Surveillance Program for Equine Influenza Virus in the United States during 2008–2021"

_pathogens, 2023, doi:10.3390/pathogens12020192_

Round 1

Reviewer 1 Report

Equine respiratory infections are one of the main causes of systemic disease, but may also result in poor-performance and significant economic loses. As such, the authors address an important subject providing insight into risk factors for EIV infection whilst also discussing the possibility of co-infection by other respiratory pathogens. Most of the risk factors presented/discussed throughout the manuscript have already been described in the other works, nonetheless the screening for co-infections raises an interesting issue. Overall the manuscript is well-written but some small issues should be addressed.

Line 34 - Influenza A/ equine/ Miami/ 63 (H3N8) – remove space after “/”.

Line 71 – The authors should considered providing the questionnaire used for obtaining signalment history information and clinical signs.

Line 81 to 86 – Upon collection did you establish a time limit for samples to be processed?

Line 84 - … “blood sample was collected in an EDTA purple top blood tube”. Please indicate blood volume. Also, if the sample was collected in a pink or white top EDTA tube would it be impossible to run the qPCR test (or would it interfere)? If it does not interfere, please consider removing “purple top blood tube” from the sentence and including if it was K3EDTA or K2EDTA or other.

Line 123 – There is a “.” missing

Line 134 – There is no subtitle for figure 1. Please correct.

Line 137 to 139 and 140 – Please indicate the name of the States and not only their abbreviation.

Line 160 – These values refer to Odds Ratio or adjusted Odds Ratio?

Line 165 – Figure 2 has no subtitle. Please correct.

Line 187 – Authors state that vaccination status influenced the number of EIV-positive cases. How many horses received a multi-agent vaccine for both EHV-1 and 4 as well as EIV? The authors refer this information might be pertinent but do not supply additional data.

Line 190 – Vaccination status “unknown” is not presented in table 3.

Line 197 – Please provide a subtitle for figure 3

Line 202 – Please correct “101.5o F (38.6oC)”

Line 207 to 217 – Since the information here described is essentially the same has the one presented in table 3, please consider reformulation this paragraph so that the same information is not repeated twice.

Line 226 – Table 5 is not mentioned in the text

Line 241 – Please correct “ > 101.50 F (38.60C)”

Line 249 to 253 – The authors address the possibility of a tri-annual vaccination protocol or the possibility of a third booster dose. Would this recommendation be in specific circumstances or applied to all equids? Also, what other data have you used to support this information? You have included a reference pertaining to the proceeding of the AAEP (2019), but I did not encounter any information which might substantiate your hypothesis.

Line 307 to 308 – “… kennel cough in dogs [18] and 307 bovine respiratory disease complex in cattle [19]…”. Consider changing to “… canine infectious tracheobronchitis (“kennel cough”) [18] and  bovine respiratory disease complex [19]…”

Line 311 to 315 – Lack of references that support this information.

In addition to the above suggestions please revise the “Results” chapter, since there are several times when the information presented in the tables is repeated again (and in the exact same way) in the text. Please consider citing in the text only information which complements the tables or that highlights certain results. If the information presented in the tables is exactly the same has the one described in the text.

Also, I would have liked the authors to discuss a bit more the implications of their findings in the discussion to better enhance the importance of their findings.

Author Response

Equine respiratory infections are one of the main causes of systemic disease, but may also result in poor-performance and significant economic loses. As such, the authors address an important subject providing insight into risk factors for EIV infection whilst also discussing the possibility of co-infection by other respiratory pathogens. Most of the risk factors presented/discussed throughout the manuscript have already been described in the other works, nonetheless the screening for co-infections raises an interesting issue. Overall the manuscript is well-written but some small issues should be addressed.

Response: Thank you for the kind comments. This paper is specifically focused from works looking at broader portions of this data but has the goal to intently focus on EIV and the uniqueness of this disease hallmarked in this dataset. The dataset has displayed the challenges including co-infections can bring when reviewing the co-association with EIV and gamma herpesviruses. EHV-2 and EHV-5 are very common in many infectious respiratory illnesses and thus present challenges with interpretation. Other more common pathogens like EHV-1/4 and S. equi are easier to define and delineate differences. I am hopeful this has been well address in the manuscript.

Line 34 - Influenza A/ equine/ Miami/ 63 (H3N8) – remove space after “/”.

Response: completed

Line 71 – The authors should considered providing the questionnaire used for obtaining signalment history information and clinical signs.

Questionnaire has been added to methods section pending layout team can make it work after first paragraph.

Line 81 to 86 – Upon collection did you establish a time limit for samples to be processed?

Response: Samples were tested within 7 days of collection. (Added to paragraph)

Line 84 - … “blood sample was collected in an EDTA purple top blood tube”. Please indicate blood volume. Also, if the sample was collected in a pink or white top EDTA tube would it be impossible to run the qPCR test (or would it interfere)? If it does not interfere, please consider removing “purple top blood tube” from the sentence and including if it was K3EDTA or K2EDTA or other.

Response: Confirmation from the laboratory- EDTA purple top blood tube is the only sample they can run. The lab cannot utilize pink or white top EDTA blood tubes.

Line 123 – There is a “.” Missing

Response: rephrased lines 122-124

Line 134 – There is no subtitle for figure 1. Please correct.

Response: Subtitle added

Line 137 to 139 and 140 – Please indicate the name of the States and not only their abbreviation.

Response: completed

Line 160 – These values refer to Odds Ratio or adjusted Odds Ratio?

Response: Confirm with Kaitlyn

Line 165 – Figure 2 has no subtitle. Please correct.

Response: completed

Line 187 – Authors state that vaccination status influenced the number of EIV-positive cases. How many horses received a multi-agent vaccine for both EHV-1 and 4 as well as EIV? The authors refer this information might be pertinent but do not supply additional data.

 Ask Kaitlyn Response: Is there a way to discern this info ?

Line 190 – Vaccination status “unknown” is not presented in table 3.

Response: The unknown vaccination values have been reviewed and added to the table.

Line 197 – Please provide a subtitle for figure 3

Response: completed

Line 202 – Please correct “101.5o F (38.6oC)”

Response: completed

Line 207 to 217 – Since the information here described is essentially the same has the one presented in table 3, please consider reformulation this paragraph so that the same information is not repeated twice.

Response: In writing this paper, I learned more about myself and many other veterinarians that walk in my shoes. Understanding tables, charts, graphs and statistical data in publications can be quite challenging and there are times when someone takes the time to right a narrative and walk me through large tables, I develop the understanding the author desired. Without the narrative support, I would have not received the impactful information I sought to learn. I am hopeful you will honor this request and keep the narrative in place as it is- for me it has been quite helpful.

Line 226 – Table 5 is not mentioned in the text

Response: Great observation and comments have been added

Line 241 – Please correct “ > 101.50 F (38.60C)”

Response: completed

Line 249 to 253 – The authors address the possibility of a tri-annual vaccination protocol or the possibility of a third booster dose. Would this recommendation be in specific circumstances or applied to all equids? Also, what other data have you used to support this information? You have included a reference pertaining to the proceeding of the AAEP (2019), but I did not encounter any information which might substantiate your hypothesis.

Response: I have added an additional sentence to qualify use of a potential 3rd EIV booster in one calendar year in horses at high levels of training where the potential loss to participate in an event is too great.

Line 307 to 308 – “… kennel cough in dogs [18] and 307 bovine respiratory disease complex in cattle [19]…”. Consider changing to “… canine infectious tracheobronchitis (“kennel cough”) [18] and  bovine respiratory disease complex [19]…”

Response: completed

Line 311 to 315 – Lack of references that support this information.

In addition to the above suggestions please revise the “Results” chapter, since there are several times when the information presented in the tables is repeated again (and in the exact same way) in the text. Please consider citing in the text only information which complements the tables or that highlights certain results. If the information presented in the tables is exactly the same has the one described in the text.

Response: With collaborating conversation with other co-authors, there was agreement that current method of presenting data in an extended narrative method closely reviewing the data was beneficial to the reader down the road in correctly interpreting the information in the tables and figures. My apologies for the repetitive method but seems helpful breaking the information into more palatable pieces for a better understanding. Some readers will lean more heavily into the narrative while others will depend upon tables, figures and charts depending on their preferred review method.

Also, I would have liked the authors to discuss a bit more the implications of their findings in the discussion to better enhance the importance of their findings.

Reviewer 2 Report

This is a very informative article on the incidence of EIV infections with and without co-infection by other viruses causing respiratory horse disease. Please find my comments and suggestions in the attached manuscript.

One major concern: The discussion somewhat lacks scientifical depth. This should be improved.  

Author Response

Responses to reviewer 2 are included as replies to each comment on the pdf version

Round 2

Reviewer 1 Report

The authors have performed most of the suggested corrections and provided suitable explanations.

Although I understand the authors reasoning about the chosen format for presenting the study's results I would like to reinforce a previous made suggestion for future manuscripts. Although it is very tempting to repeat in the narrative exactly the same information that is shown in the figures or tables, it can also mean that the latter are unecessary/redundanct. 

Overall the authors have done a good job.